# Exploring PGE2 and LXA4 Levels in Migraine Patients: The Potential of LXA4-Based Therapies

**DOI:** 10.3390/diagnostics14060635

**Published:** 2024-03-17

**Authors:** Idris Kocaturk, Sedat Gulten, Bunyamin Ece, Fatma Mutlu Kukul Guven

**Affiliations:** 1Department of Neurology, Kastamonu University, Kastamonu 37150, Türkiye; 2Department of Biochemistry, Kastamonu University, Kastamonu 37150, Türkiye; sedat_gulten@hotmail.com; 3Department of Radiology, Kastamonu University, Kastamonu 37150, Türkiye; bunyaminece@kastamonu.edu.tr; 4Department of Emergency Medicine, Kastamonu University, Kastamonu 37150, Türkiye; mutlukukul@kastamonu.edu.tr

**Keywords:** migraine, inflammation, prostaglandin E2, lipoxin A4, C-reactive protein, fibrinogen

## Abstract

Neurogenic inflammation plays a significant role in the pathogenesis of migraines. This study aimed to investigate the serum levels of prostaglandin E2 (PGE2), lipoxin A4 (LXA4), and other inflammatory biomarkers (C-reactive protein, fibrinogen) in migraine patients. In total, 53 migraine patients and 53 healthy controls were evaluated. Blood serum samples were collected during both attack and interictal periods and compared with the control group. In both the attack and interictal periods, PGE2 and LXA4 values were significantly lower in migraine patients compared to the control group (*p* < 0.001). Additionally, PGE2 values during the attack period were significantly higher than those during the interictal period (*p* = 0.016). Patients experiencing migraine attacks lasting ≥ 12 h had significantly lower serum PGE2 and LXA4 levels compared to those with attacks lasting < 12 h (*p* = 0.028 and *p* = 0.009, respectively). In ROC analysis, cut-off values of 332.7 pg/mL for PGE2 and 27.2 ng/mL for LXA4 were determined with 70–80% sensitivity and specificity. In conclusion, PGE2 and LXA4 levels are significantly lower in migraine patients during both interictal and attack periods. Additionally, the levels of LXA4 and PGE2 decrease more with the prolongation of migraine attack duration. Our findings provide a basis for future treatment planning.

## 1. Introduction

Migraines are a common neurological disorder with a worldwide prevalence ranging from 14 to 15%, affecting quality of life [1]. Recent studies have investigated the mechanisms underlying migraines and have identified neurogenic inflammation surrounding dural trigeminal afferents as a crucial factor in the development of migraine attacks [2]. Trigeminovascular pathophysiology has been widely accepted for many years in migraines [3,4]. This theory posits that trigeminal nerve stimulation leads to vasodilation and neurogenic inflammation. Neurogenic inflammation causes vascular permeability to increase, resulting in leukocyte infiltration and glial cell activation. These changes lead to the production of inflammatory mediators, including cytokines and chemokines. It also increases blood–brain barrier permeability. Mast cell degranulation releases neuropeptides through trigeminal nerve activation. It can also activate and sensitize nociceptors through hormonal fluctuations or cortical spreading depression [5,6]. There are studies investigating the role of molecules such as C-reactive protein (CRP), fibrinogen, and prostaglandins (PGs) in the pathophysiology of neurogenic inflammation in migraines [7,8,9].

The presence of higher levels of CRP in migraine patients indicates a link between inflammation and pathogenesis [7]. Fibrinogen is also crucial in systemic inflammatory responses. Like CRP, high levels of fibrinogen have been associated with migraines, supporting the role of inflammation in migraine development [10].

PGs, members of the eicosanoid family, are produced by nearly all cells in the body. Prostaglandin E2 (PGE2), considered the most crucial human prostanoid, demonstrates homeostatic, inflammatory, and anti-inflammatory properties [11,12]. PGE2 plays a role in the underlying mechanisms of pain in migraines [12]. In studies, PGE2 levels were compared in migraine patients during the attack and interictal periods, revealing higher levels during the attack period than in the interictal period [13]. However, to our knowledge, there is no study in the literature investigating PGE2 levels in three groups, including the attack period, interictal period, and healthy controls.

Lipoxins (LXs) are bioactive molecules derived from lipids and synthesized from arachidonic acid (AA). They exhibit potent anti-inflammatory and pro-resolving activities [14]. Inflammation resolution is a dynamic process actively regulated by specialized pro-resolving lipid mediators (SPMs) [15]. LXs are lipid-based signaling molecules that play a crucial role in resolving inflammation [14]. They are produced from AA in platelets and neutrophils via two main pathways. LXs act through four G protein-coupled receptors (GPCRs), including ALX/FPR2 [14]. ALX/FPR2 activation decreases neutrophil recruitment, enhances anti-inflammatory factor production, and facilitates apoptotic leukocyte phagocytosis [16]. Lipoxin A4 (LXA4) is the most potent lipoxin, exhibiting anti-inflammatory and neuroprotective effects [17,18,19]. The link between LXA4 and several neurological disorders, including stroke and multiple sclerosis, has been evaluated [20]. However, we could not find any research in the literature that examined LXA4 levels in migraine patients and their link to migraine characteristics. In this regard, our study will be the first in the field to assess LXA4 levels in migraine patients.

The aim of this study is to examine differences in serum levels of inflammatory molecules, including PGE2, LXA4, CRP, and fibrinogen, between migraine patients and healthy controls, as well as whether these molecules have associations with the clinical features of migraines.

## 2. Materials and Methods

Ethics committee approval was obtained from the hospital’s local ethics committee for this prospective study (ethics committee no: 2023-KAEK-83, decision date: 7 May 2023). Informed consent was obtained from all individuals participating in this study. This study was conducted in accordance with the Declaration of Helsinki. For our study, migraine patients who presented to the emergency department due to a migraine attack between 15 July 2023 and 15 September 2023 were evaluated.

The inclusion criteria for these evaluated migraine patients were being a voluntary participant in this study, being between the ages of 18 and 50, having a confirmed diagnosis of episodic migraines according to the International Classification of Headache Disorders (third edition) (ICHD-3) [21], having no medical history other than migraines, having a body mass index (BMI) below 30 kg/m^2^, not currently using any medications, not currently using any prophylactic migraine medication, and not taking any anti-migraine medication for at least one week before giving a blood sample.

The exclusion criteria for these evaluated migraine patients were experiencing persistent or recurrent headaches during the week following a migraine attack and being diagnosed with a treatment-resistant migraine attack.

Out of the 101 migraine patients who presented to the emergency department within the specified date range, 53 migraine patients who met the inclusion and exclusion criteria were included in this study (Figure 1).

The control group comprised 53 healthy volunteers aged 18–50 with a BMI below 30 and no medical history or drug usage.

Blood samples were taken from the 53 migraine patients in the emergency department. After blood collection, Diclofenac sodium (75 mg) was administered intramuscularly as a treatment [22]. Blood samples were collected again from the same patients after at least seven days following the cessation of migraine attacks when they were called for a follow-up appointment at our hospital’s neurology outpatient clinic to evaluate the interictal period. Blood samples were collected from the control group in the morning after an overnight fast. The gender, age, weight, and height of these individuals were recorded. All samples collected from the migraine patients at the attack and interictal periods and collected from the control group were promptly centrifuged and stored at −20 °C.

In addition to demographic data such as age and gender, characteristics such as family history, presence of aura, migraine duration, attack frequency, attack duration, presence of white matter lesion on brain MRI, and MIDAS scores were recorded from the migraine group patients. Based on factors such as headache frequency (≤4 and >4 per month), attack duration (<12 h and ≥12 h), and disease duration (≤5 and >five years), patients were divided into two groups. To improve the statistical analysis power in the MIDAS subgroup, we also specified two categories (mild and severe) instead of four grades (minimum, mild, moderate, and severe). Blood data were obtained from migraine patients during the attack and interictal periods and were obtained from control group patients.

### 2.1. Laboratory Tests Analyzes

Blood samples were collected from migraine patients in the emergency room within the first four hours of the onset of their headaches. For the interictal and control groups, morning fasting blood samples were preferred. Serum LXA4, PGE2, CRP, and plasma fibrinogen levels were measured from these blood samples. After blood collection, samples underwent centrifugation at 2000× *g* for 10 min at 25 °C. The separated serum was aliquoted and cryopreserved at −20 °C for up to one month. Serum LXA4 and PGE2 levels were measured using the ELISA method (Gentaur, Kampenhout, Belgium (catalog number 576-201-12-0613) for LXA4, and Sunredbio, Shanghai, China (catalog number 576-201-12-1010) for PGE2). Plasma fibrinogen levels were measured using a coagulation system (CS-2500, Sysmex Corporation, Kobe, Japan), and serum CRP levels were measured using a biochemistry system (AU5800 Analyzer, Beckman Coulter, Brea, CA, USA).

### 2.2. Imaging Procedure

Migraine patients who had not had a brain MRI in the previous year were scanned with a 1.5T MRI scanner (SIGNA Explorer, GE Healthcare, Waukesha, WI, USA). The MRI protocol included T1-weighted, T2-weighted, and fluid-attenuated inversion recovery (FLAIR) sequences to provide detailed anatomical information about the brain. MRI scans were examined to identify white matter lesions (WMLs) with high signal intensity on T2-weighted and FLAIR images. Migraine patients were divided into two groups based on the presence of WMLs on MRI: Group 1, consisting of participants without any detectable WMLs, and Group 2, which consisted of participants with hyperintense lesions on MRI.

### 2.3. Statistical Analysis

The data were analyzed using the statistical software package IBM SPSS, version 18.0, to assess the dataset’s characteristics. To summarize categorical variables, frequencies and percentages were calculated, while for numerical variables, medians along with the 25th and 75th percentiles were determined. The normality of data distribution was assessed using the Kolmogorov–Smirnov test. A Continuity Correction test was employed to investigate potential differences in gender distribution between the groups. To visualize the data distribution, boxplots were created. Due to non-normality and independent groups, the Mann–Whitney U test was used to compare the control and attack and control and interictal groups. The Wilcoxon signed-rank test was used to compare migraine patients in the interictal period and attack period, taking into account non-normality and paired data. To assess the levels of PGE2 and LXA4 diagnostic efficacy, we performed an ROC analysis between interictal-period migraine patients and healthy controls. The area under curve (AUC) was calculated using Youden’s index, reflecting the biomarker’s overall accuracy. We also determined the cut-off value, sensitivity, and specificity according to Youden’s index. While investigating the associations between non-normally distributed and/or ordinal variables, the correlation coefficients and their significance were calculated using the Spearmen test. A *p*-value of <0.05 was considered statistically significant.

## 3. Results

In total, 53 migraine patients (11 male, 42 female) and 53 healthy control individuals (16 male, 37 female) were included in this study. The mean age in the migraine group was 37.3 ± 9.2 years, and the mean age in the healthy control group was 34.9 ± 8.9 years. No significant differences were found between the migraine patient group and the control group in terms of mean ages and gender distributions (respectively, *p* = 0.540 and *p* = 0.373). The anamnestic, clinical, and imaging findings of the migraine patient group are shown in Table 1.

In the comparison of serum PGE2, LXA4, CRP, and fibrinogen values obtained during the attack and interictal periods in migraine patients with the healthy control group, during both the attack and interictal periods, serum PGE2 and LXA4 values were statistically significantly lower in migraine patients compared to the healthy control group (*p* < 0.001) (Table 2, Figure 2). However, no significant difference was observed in fibrinogen and CRP values between the groups (*p* > 0.05) (Table 2, Figure 2).

In migraine patients, while no significant differences were found in serum LXA4, fibrinogen, and CRP values during the attack and interictal periods, the serum PGE2 value was statistically significantly lower during the interictal period compared to the attack period (*p* = 0.016) (Table 2, Figure 2).

In the comparison of PGE2 and LXA4 levels based on clinical, anamnestic, and imaging findings during the interictal period in patients, individuals experiencing migraine attacks lasting 12 h or longer showed significantly lower serum PGE2 and LXA4 levels compared to those with attacks lasting less than 12 h (*p* = 0.028 and *p* = 0.009, respectively) (Table 3). No statistically significant differences were found among other clinical characteristics (*p* > 0.05) (Table 3). Additionally, the correlation analysis revealed a reverse correlation between attack duration and levels of LXA4 and PGE2 (ρ = −0.310, *p* = 0.024 and ρ = −0.304, *p* = 0.027, respectively).

The results of the ROC analysis for the PGE2 and LXA4 values obtained from migraine patients and normal control groups are shown in Table 4. Accordingly, cut-off values of 332.7 pg/mL (with 74% sensitivity and 81% specificity) for PGE2 and 27.2 ng/mL (with 81% sensitivity and 70% specificity) for LXA4 were determined (Table 4, Figure 3).

## 4. Discussion

The most notable findings of our study were that PGE2 and LXA4 levels were significantly lower in migraine patients than in healthy controls in both the attack and interictal periods. Furthermore, PGE2 and LXA4 levels were found to be significantly lower in migraine patients who had attacks lasting more than 12 h than in patients who had attacks lasting less than 12 h. There was no significant difference in CRP and fibrinogen levels between migraine patients and healthy controls.

LXA4 has been shown to have anti-inflammatory and neuroprotective effects in a variety of neurological diseases [23,24]. Mice lacking the ALX/FPR2 homolog develop a severe inflammatory response after reperfusion in cerebral ischemia [25]. LXA4 reduces neurogenic damage and promotes neuroprotection in ischemic stroke [25]. Studies in rats have shown that LXA4 injections into the ischemic brain tissue reduce stroke volume [20]. LXA4 also positively affects ischemic stroke by correcting the blood–brain barrier (BBB) dysfunction that develops after stroke [26,27]. Moreover, a recent study showed that SPMs, including LXA4, are reduced in multiple sclerosis (MS) patients [24]. Lipid metabolites in plasma patients of MS correlate with disease progression, and SPM production is diminished in leukocytes [28].

The most commonly used molecules in acute and chronic pain are NSAIDs, opioids, and corticosteroids. However, their use has problems such as side effects, addiction, and tolerance [29]. Isolated SPMs have been used as analgesics in experimental models at very low doses without side effects [30]. Moreover, it has been shown that diets containing eicosapentaenoic acid and docosahexaenoic acid, which are precursors of SPMs, reduce the frequency and duration of migraine attacks [31]. LXA4 is endogenous in all SPMs and has analgesic effects. Intravenous or intrathecal injections of LXA4 reduced thermal hyperalgesia in rats [32]. Miao et al. showed that LXA4 effectively reduced radicular pain in a rat non-compressive lumbar disc herniation model [33]. Martini et al. also revealed that LXA4 may efficiently regulate microglial activation and reduce neuropathic pain in animals following spinal cord hemisection [34]. In our study, LXA4 was significantly lower in migraine patients than in the healthy group during the attack and interictal periods. This result is striking and, as far as we know, is not included in the literature. Decreased LXA4 levels suggest that migraine patients may have impaired LXA4 production or catabolism. However, the literature cannot provide sufficient information on this subject. LXA4 may be a new and promising agent, and has the potential to help resolve inflammation and shorten the duration of migraine attacks. While these findings are promising, additional research is needed to confirm their validity and to develop safe and effective LXA4-based therapies [35].

PGs are signaling molecules produced by cyclooxygenase (COX) enzymes from AA. PGE2 is the most common prostaglandin in the human body. PGE2 receptors are a family of G protein-coupled receptors (GPCRs), a lipid mediator in inflammation, pain, and immunity [36]. PGE2 levels have been linked to a variety of neurological diseases, including Alzheimer’s disease, multiple sclerosis, and migraines. In Alzheimer’s patients, PGE2 levels are high [36]. PGE2 appears to have a protective effect in multiple sclerosis by activating myelin-producing oligodendrocytes [37]. PGE2 is also implicated in the mechanisms underlying migraine pain [38]. Several studies demonstrated that the infusion of PGE2 causes headaches in 83–100% of patients, but the headache occurrence rate after prostaglandin F2-alpha infusion was just 17% [39,40,41]. Internal jugular venous blood levels of PGE2 have been found by Sarchielli et al. to peak two hours after the onset of a headache, remain throughout hours 4–6 post-headache start, and then drop [42]. PGE2 levels in migraine patients’ saliva and nasal lavage samples taken during the interictal stage are similar to those in control subjects, according to research by Mohammadian and colleagues [43]. Tuca et al. found that PGE2 levels in migraine sufferers’ saliva are considerably greater during a headache attack than in samples obtained between attacks [8]. A recent study analyzing COX-2 levels in migraine patients discovered a statistically significant upregulation of this PGE2-producing enzyme during headache episodes [44]. In our study, PGE2 levels were demonstrated to be higher in migraine patients during the attack period than in the interictal period, which is consistent with previous research. Furthermore, it was significantly lower than the healthy control group during the attack and interictal periods. This finding could imply that the role of PGE2 in migraine pain and the pathophysiological pathways are more complex than previously understood. More detailed research is required on this topic.

The relevance of CRP in migraine pathogenesis is still unknown, and various studies on the subject have yielded conflicting results [45]. While some studies observed elevated CRP levels in migraine patients, others showed no significant differences [46]. Researchers believe that body mass index (BMI) may play an important influence. Studies on overweight or obese subjects consistently revealed higher levels of CRP in migraineurs [7,46,47]. Conversely, studies with normal-weight or underweight participants generally found no significant CRP differences between migraineurs and healthy controls [7,48]. Since we did not include obese individuals in this study, we may not have been able to detect a significant relationship between CRP levels during attack, interictal periods, and healthy controls.

Some studies have shown that fibrinogen, a protein involved in blood clotting and inflammation, may potentially have a role in migraines [10]. Research investigating the association between fibrinogen and migraines has produced conflicting findings, with some studies indicating lower fibrinogen levels in migraine patients while others report elevated levels [10,49]. Additionally, a population-based study suggested an association between high fibrinogen levels and increased migraine attack frequency in women [49]. No significant difference was observed in fibrinogen levels between groups in our study.

There are several limitations to our investigation. The size of our study group is not excessively large. To provide enough statistical power, we divided the disease’s duration, the attack’s duration, and the frequency into subgroups. Moreover, the four standard ratings for the MIDAS subgroup—minimum, mild, moderate, and severe—were decreased to just two (mild and severe). We were unable to establish a causal relationship between peripheral blood levels of CRP, fibrinogen, PGE2, and LXA4 and migraine because of the case–control design of our investigation. Furthermore, there were a lot of confounding variables that we could have controlled for that could have affected the marker levels. For example, when blood samples were taken from migraine patients during episodes, the fasting requirement was not fulfilled. Furthermore, it was not possible to determine the amounts of the studied biomarkers in cerebrospinal fluid (CSF) at the same time. Consequently, it should be considered that the peripheral biomarker levels in our study participants may not accurately reflect changes in their central nervous systems.

## 5. Conclusions

In conclusion, our study revealed lower serum PGE2 and LXA4 levels in migraine patients compared to healthy individuals. Moreover, our results showed a relationship between decreased PGE2 and LXA4 levels and the length of migraine attacks. Reduced levels of LXA4, a key mediator in inflammation resolution, were connected to prolonged inflammation and attack durations. Thus, LXA4 may be a promising target in migraine treatment. Future studies with more cases are required to investigate this issue further.

## Figures and Tables

**Figure 1 diagnostics-14-00635-f001:**
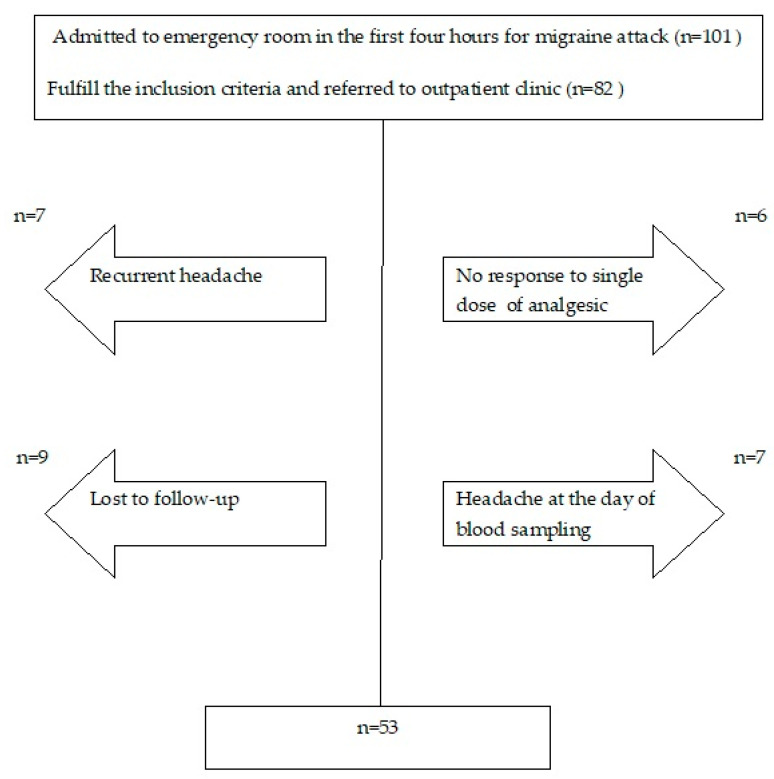
Flowchart of patient selection.

**Figure 2 diagnostics-14-00635-f002:**
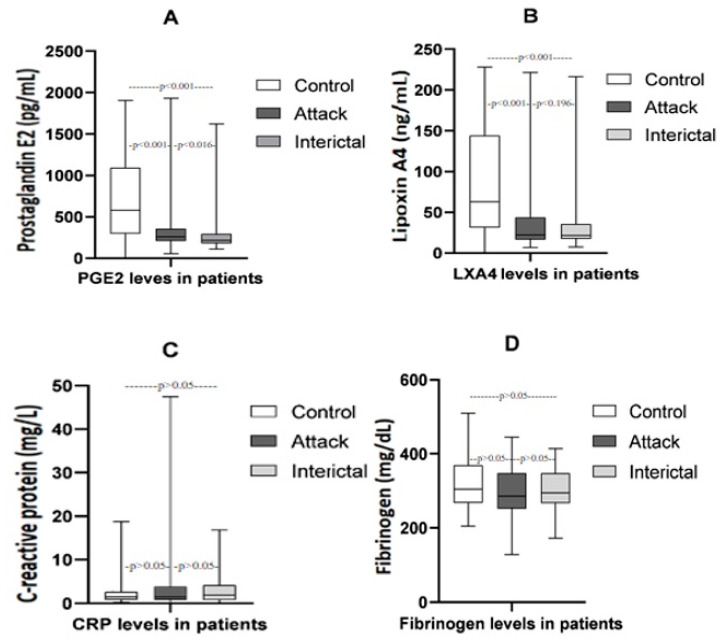
Comparison of serum concentrations of PGE2, LXA4, CRP, and fibrinogen in patients with migraines. The figure shows that serum PGE2 (**A**) and LXA4 (**B**) values in migraine patients, both attack and interictal groups, are statistically significantly lower than in the healthy control group. The medians and interquartile ranges (IQRs) of serum levels of LXA4, PGE2, CRP (**C**), and fibrinogen (**D**) for all groups are shown using box plots.

**Figure 3 diagnostics-14-00635-f003:**
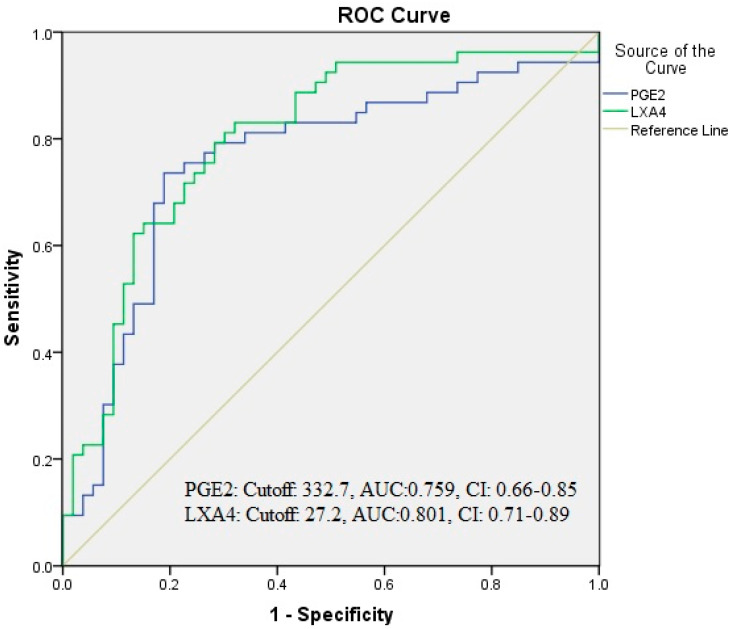
Receiver operating characteristic (ROC) curve analysis of interictal migraine patients.

**Table 1 diagnostics-14-00635-t001:** The anamnestic, clinical, and imaging findings of the migraine patient group.

Migraine Group	*n* (%)
Family history	
positive	31 (58%)
negative	22 (42%)
Aura	
with	21 (40%)
without	32 (60%)
Disease duration (years)	
≤5	26 (49%)
>5	27 (51%)
Headache frequency (per month)	
≤4	21 (40%)
>4	32 (60%)
Attack duration (hours)	
<12	21 (40%)
≥12	32 (60%)
White matter lesions	
Positive	17 (32%)
Negative	36 (68%)
MIDAS score	
Mild	38 (72%)
Severe	15 (28%)
Total	53 (100)

MIDAS: Migraine Disability Assessment Score.

**Table 2 diagnostics-14-00635-t002:** Comparison of PGE2, LXA4, fibrinogen, and CRP values in migraine patients and healthy controls.

	Migraine Attack Group ^1^ Median (Q1;Q3)	Migraine Interictal Group ^2^ Median (Q1;Q3)	Healthy Control Group ^3^ Median (Q1;Q3)	^1–2^ *p* *	^1–3^ *p* *	^2–3^ *p* *
PGE2, pg/mL	257 (207; 358)	218 (180; 296)	577 (294; 1094)	0.016	<0.001	<0.001
LXA4, ng/mL	22.1 (16.4; 43.6)	21.8 (17.3; 35.6)	63.1 (31.1; 144)	0.196	<0.001	<0.001
Fibrinogen, g/L	285 (251; 348)	294 (266; 348)	304 (267; 370)	0.557	0.199	0.150
CRP, mg/dL	1.54 (0.78; 3.9)	1.95 (0.73; 4.22)	1.45 (0.80; 2.7)	0.538	0.558	0.558

* Mann–Whitney U Test, PGE2: prostaglandin E2, LXA4: lipoxin A4, CRP: C-reactive protein, Q: quartile, ^1^: migraine attack group, ^2^: migraine interictal group, ^3^: healthy control group.

**Table 3 diagnostics-14-00635-t003:** Comparison of PGE2 and LXA4 levels based on clinical, anamnestic, and imaging findings during the interictal period in patients with migraines.

		PGE2, pg/mL		LXA4, ng/mL	
Migraine Interictal Group	n (%)	Median (Q1; Q3)	*p* Value	Median (Q1; Q3)	*p* Value
Family history					
positive	31 (58%)	210 (179–367)	0.576	23 (18; 43)	0.386
negative	22 (42%)	236 (177–285)		20 (16; 29)	
Aura					
with	21 (40%)	212 (180–287)	0.827	22 (18–36)	0.461
without	32 (60%)	222 (180–303)		19 (16–37)	
Disease duration (years)					
≤5	26 (49%)	211 (179–317)	0.722	20 (16–43)	0.783
>5	27 (51%)	220 (179–292)		22 (17–30)	
Headache frequency (per month)					
≤4	21 (40%)	212 (173–262)	0.434	22 (16–30)	0.519
>4	32 (60%)	219 (182–351)		21 (17–40)	
Attack duration (hours)					
<12	21 (40%)	241 (206–477)	0.028	25 (20–54)	0.009
≥12	32 (60%)	207 (166–248)		19 (16–25)	
White matter lesions					
Positive	17 (32%)	231 (205–318)	0.230	25 (19–40)	0.286
Negative	36 (68%)	209 (168–270)		19 (16–30)	
MIDAS score					
Mild	38 (72%)	209 (178–240)	0.710	20 (16–31)	0.248
Severe	15 (28%)	276 (192–589)		25 (18–44)	
Total	53 (100)				

**Table 4 diagnostics-14-00635-t004:** ROC analysis for the PGE2 and LXA4 values.

	PGE2, pg/mL	LXA4, ng/mL
Cut-off Value	332.7	27.2
Area Under the ROC Curve (95% Confidence Interval)	0.759 (0.66–0.85)	0.801 (0.71–0.89)
*p* Value	<0.001	<0.001
Sensitivity	74%	81%
Specificity	81%	70%
Positive Predictive Value	80%	73%
Negative Predictive Value	77%	79%

PGE2: prostaglandin E2, LXA4: lipoxin A4.

## Data Availability

The data presented in this study are available on request from the corresponding author. The data are not publicly available due to issues of patient privacy.

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
