# Peer review of "Exploring PGE2 and LXA4 Levels in Migraine Patients: The Potential of LXA4-Based Therapies"

_diagnostics, 2024, doi:10.3390/diagnostics14060635_

Round 1
Reviewer 1 Report
Comments and Suggestions for Authors
The manuscript “Exploring PGE2 and LXA4 Levels in Migraine Patients: The Potential of LXA4-Based Therapies“ by Idris Kocaturk et al. is a research article investigating the role of inflammatory biomarkers in the pathogenesis of migraine. The experimental design compares migraine attacks and interictal periods with the control group. The manuscript is well organized. Methods and statistical analyses are appropriate, and the results are presented appropriately. The role of PGE2 in migraine was quite often investigated but data on the role of LXA4 are very scarce. This gives an innovative character to the manuscript. The references are adequate. Since 25 of the 43 references are older than ten years, it might be possible to add a few newer ones in addition to the existing ones.
These are some suggestions to authors that I hope could contribute to the quality of the manuscript:
Section 2 – Materials and Methods
- Page 2, line 82: Add the reference for ICHD-3 and adjust the reference list accordingly.
- Page 4, line 123: There is a catalogue number for LXA4 serum ELISA kit. The catalogue number for PgE2 serum ELISA kit should be added.
- Page 4, line 129: The manufacturer for GE SIGNA Explorer has to be added.
Section 3 - Results
- Page 4, lines 160-161: The word anamnesis should be replaced with the word anamnestic. The same applies to lines 163, 189, and 197.
- Page 6, Table 2: Correct Fibrinojen to Fibrinogen.
- Page 6, Figure 2 :
· The graphs within Figure 2 should be labeled as A, B, C, and D.
· Units along the ordinates (pg/ml, ng/ml…) should be vertically oriented.
· Units along the ordinates should be uniform. In Table 2, the abbreviation for liter is written as L, and in Figure 2, L and l are used.
· Graph legends could have a slightly smaller font and be closer to the corresponding graph.
· The order of experimental groups (boxes) in all 4 graphs should be the same, as indicated in the graph legend.
· The filling (shading) of the group boxes should be lighter because the median value line is barely visible on some of them.
· In the description of Figure 2, it is mentioned that there is a statistically significant difference between some groups, but it is not marked anywhere on the graphs. Mark statistical significance on the graphs, and in the description of the figure write p values with the appropriate marks used.
- Page 6, Figure 2, lines 177-180 – In the description, after Figure 2. should be a sentence which is “an image title" : Comparison of serum concentrations of PGE2, LXA4, CRP, and fibrinogen in patients with migraine. This should be followed by the text that was previously written after Figure 2.: “The figure shows that serum PGE2 (A) and LXA4 (B) values in migraine patients, both attack and interictal groups are statistically significantly lower than in the healthy control group. The medians and interquartile ranges (IQRs) of serum levels of LXA4, PGE2, CRP (C), and fibrinogen (D) for all groups are shown using box plots.“ The suggested labels A, B, C, and D are incorporated into the text.
Section 4 – Discussion
- Pages 9-10, lines 270-278: Reference 42 is missing from the text, although it is in the reference list.
Author Response
Manuscript ID: diagnostics-2890151
Exploring PGE2 and LXA4 Levels in Migraine Patients: The Potential of
LXA4-Based Therapies
|
|
|||||||||||||||||||||||||||||||||||||
|
1. Summary |
|
|
|||||||||||||||||||||||||||||||||||
|
Dear Reviewer, We appreciate the time and effort you dedicated to providing feedback on our manuscript and are grateful for the insightful comments and valuable improvements to our paper. We have made the relevant corrections according to your suggestions. Please see below for a point-by-point response to the comments and suggestions. Please see the attachment for the last edited version of the article with track changes enabled in the format Word file.
2. Questions for General Evaluation Open Review (x) I would not like to sign my review report Quality of English Language ( ) I am not qualified to assess the quality of English in this paper
|
|||||||||||||||||||||||||||||||||||||
|
3. Point-by-point response to Comments and Suggestions for Authors The manuscript “Exploring PGE2 and LXA4 Levels in Migraine Patients: The Potential of LXA4-Based Therapies“ by Idris Kocaturk et al. is a research article investigating the role of inflammatory biomarkers in the pathogenesis of migraine. The experimental design compares migraine attacks and interictal periods with the control group. The manuscript is well organized. Methods and statistical analyses are appropriate, and the results are presented appropriately. The role of PGE2 in migraine was quite often investigated but data on the role of LXA4 are very scarce. This gives an innovative character to the manuscript.
|
|||||||||||||||||||||||||||||||||||||
|
Comments 1: The references are adequate. Since 25 of the 43 references are older than ten years, it might be possible to add a few newer ones in addition to the existing ones. |
|||||||||||||||||||||||||||||||||||||
|
Response 1: Thank you for pointing this out. As you suggested, a few newer ones in addition to the existing ones. Added references are listed below.
3. Ramachandran R. Neurogenic inflammation and its role in migraine. Semin Immunopathol. 2018;40(3):301-314.
12. Ramsden CE, Zamora D, Faurot KR, et al. Dietary alteration of n-3 and n-6 fatty acids for headache reduction in adults with migraine: randomized controlled trial. BMJ. 2021;374:n1448. Published 2021 Jun 30.
15. Kahnt AS, Schebb NH, Steinhilber D. Formation of lipoxins and resolvins in human leukocytes. Prostaglandins Other Lipid Mediat. 2023;166:106726.
31. Johansson JU, Woodling NS, Shi J, Andreasson KI. Inflammatory Cyclooxygenase Activity and PGE2 Signaling in Models of Alzheimer's Disease. Curr Immunol Rev. 2015;11(2):125-131.
42. Yazar HO, Yazar T, Aygün A, Kaygisiz Ş, Kirbaş D. Evaluation of simple inflammatory blood parameters in patients with migraine. Ir J Med Sci. 2020;189(2):677-683.
Section 2 – Materials and Methods |
|||||||||||||||||||||||||||||||||||||
|
Comments 2: Page 2, line 82: Add the reference for ICHD-3 and adjust the reference list accordingly. |
|||||||||||||||||||||||||||||||||||||
|
Response 2: In line with your suggestions, the reference for ICHD-3 was added, and the reference list was adjusted accordingly. The inclusion criteria for these evaluated migraine patients were, respectively: being a voluntary participant in the study, being between the ages of 18 and 50, having a con-firmed diagnosis of episodic migraine according to the International Classification of Headache Disorders third edition (ICHD-3) (20), 20. IHS Classification ICHD-3 The International Classification of Headache Disorders 3rd edition (Beta version) [2020 Nov 11]. Internet. Available from: https://ichd-3.org/.
Comment 3: Page 4, line 123: There is a catalogue number for LXA4 serum ELISA kit. The catalogue number for PgE2 serum ELISA kit should be added. Response 3: In line with your suggestions, the catalogue number for PgE2 serum ELISA kit was added. Serum LXA4 and PGE2 levels were measured using the ELISA method (Gentaur, Kampenhout, Belgium; catalog number 576-201-12-0613 for LXA4, and Sunredbio, Shangai, China; catalog number 576-201-12-1010 for PGE2).
Comment 4: Page 4, line 129: The manufacturer for GE SIGNA Explorer has to be added. Response 4: In line with your suggestions, the manufacturer for GE SIGNA Explorer was added. Migraine patients who had not had a brain MRI in the previous year were scanned with a 1.5T MRI scanner (SIGNA Explorer, GE Healthcare, Waukesha, WI, USA). Section 3 - Results Comment 5: Page 4, lines 160-161: The word anamnesis should be replaced with the word anamnestic. The same applies to lines 163, 189, and 197. Response 5: The word anamnesis was replaced with the word anamnestic in lines of 160, 161, 163, 189, and 197. The anamnestic, clinical, and imaging findings of the migraine patient group are shown in Table 1. The anamnestic, clinical, and imaging findings of the migraine patient group In the comparison of PGE2 and LXA4 levels based on clinical, anamnestic, and imaging findings during the interictal period in patients; individuals experiencing migraine attacks lasting 12 hours or longer showed significantly lower serum PGE2 and LXA4 levels compared to those with attacks lasting less than 12 hours (p = 0.028 and p = 0.009, respectively) (Table 3). Table 3. Comparison of PGE2 and LXA4 levels based on clinical, anamnestic, and imaging findings during the interictal period in patients with migraine
Comment 6: Page 6, Table 2: Correct Fibrinojen to Fibrinogen. Response 6: Thank you for this correction. Fibrinojen was changed to fibrinogen in Table 2.
- Comments 7: Page 6, Figure 2 : · The graphs within Figure 2 should be labeled as A, B, C, and D. · Units along the ordinates (pg/ml, ng/ml…) should be vertically oriented. · Units along the ordinates should be uniform. In Table 2, the abbreviation for liter is written as L, and in Figure 2, L and l are used. · Graph legends could have a slightly smaller font and be closer to the corresponding graph. · The order of experimental groups (boxes) in all 4 graphs should be the same, as indicated in the graph legend. · The filling (shading) of the group boxes should be lighter because the median value line is barely visible on some of them. · In the description of Figure 2, it is mentioned that there is a statistically significant difference between some groups, but it is not marked anywhere on the graphs. Mark statistical significance on the graphs, and in the description of the figure write p values with the appropriate marks used. - Page 6, Figure 2, lines 177-180 – In the description, after Figure 2. should be a sentence which is “an image title" : Comparison of serum concentrations of PGE2, LXA4, CRP, and fibrinogen in patients with migraine. This should be followed by the text that was previously written after Figure 2.: “The figure shows that serum PGE2 (A) and LXA4 (B) values in migraine patients, both attack and interictal groups are statistically significantly lower than in the healthy control group. The medians and interquartile ranges (IQRs) of serum levels of LXA4, PGE2, CRP (C), and fibrinogen (D) for all groups are shown using box plots.“ The suggested labels A, B, C, and D are incorporated into the text. Responses 7: Thank you for your valuable contribution and positive comments. Upon your request, · The graphs within Figure 2 were labeled as A, B, C, and D. · Units along the ordinates (pg/ml, ng/ml…) were vertically oriented. · Units along the ordinates were uniformed. The abbreviation for liter is written as L in Figure 2 too. · Graph legends made a slightly smaller font and made closer to the corresponding graph. · The order of experimental groups (boxes) in all 4 graphs was made same, as indicated in the graph legend. · The filling (shading) of the group boxes was made lighter. · In the description of Figure 2, statistical significance was marked on the graphs, and in the description of the figure, p values were written with the appropriate marks used. - Page 6, Figure 2, lines 177-180 – In the description, after Figure 2. a sentence which is “an image title" : Comparison of serum concentrations of PGE2, LXA4, CRP, and fibrinogen in patients with migraine. Was added. This followed by the text that was previously written after Figure 2.: “The figure shows that serum PGE2 (A) and LXA4 (B) values in migraine patients, both attack and interictal groups are statistically significantly lower than in the healthy control group. The medians and interquartile ranges (IQRs) of serum levels of LXA4, PGE2, CRP (C), and fibrinogen (D) for all groups are shown using box plots.“ The suggested labels A, B, C, and D are incorporated into the text.
Figure 2. Comparison of serum concentrations of PGE2, LXA4, CRP, and fibrinogen in patients with migraine. The figure shows that serum PGE2 (A) and LXA4 (B) values in migraine patients, both attack and interictal groups are statistically significantly lower than in the healthy control group. The medians and interquartile ranges (IQRs) of serum levels of LXA4, PGE2, CRP (C), and fibrinogen (D) for all groups are shown using box plots.
Section 4 – Discussion Comment 7: Pages 9-10, lines 270-278: Reference 42 is missing from the text, although it is in the reference list. Response 7: Thank you very much for this correction. Pages 9-10, lines 270-278: Reference 42 was added to the text. (Reference numbers have changed) Researchers believe the body mass index (BMI) may play an important influence. Studies on overweight or obese subjects consistently revealed higher levels of CRP in migraineurs [6,47,49]. (Note: Reference numbers have changed)
|
|||||||||||||||||||||||||||||||||||||

Reviewer 2 Report
Comments and Suggestions for Authors
This very interesting manuscript is devoted to exploring PGE2 and LXA4 levels in migraine patients. It was shown, what PGE2 and LXA4 levels are significantly lower in migraine patients during both interictal and attack periods. Additionally, the levels of LXA4 and PGE2 decrease more with the prolongation of migraine attack duration. However, in the Discussion section there is no discussion of possible mechanisms and reasons for such reduced level. I think that it would be add to the Discussion chapter.
In the chapter materials and methods are indicated, that after blood collection, Diclofenac sodium 75 mg was administered intramuscularly as a treatment. Were samples taken after treatment with Diclofenac to investigate PGE2 and LXA4 levels?
Author Response
Manuscript ID: diagnostics-2890151
Exploring PGE2 and LXA4 Levels in Migraine Patients: The Potential of
LXA4-Based Therapies
|
1. Summary Dear Reviewer, |
|
|
|||||||||||||||||||||||||||||||||||
|
Thank you for your valuable contribution and positive comments. Upon your request, the article has been revised and necessary corrections have been made. With all due respect, we hope our revision has satisfied you. Please find the detailed responses below and the corresponding revisions and highlighted changes in the re-submitted files. 2. Questions for General Evaluation Open Review ( ) I would not like to sign my review report Quality of English Language ( ) I am not qualified to assess the quality of English in this paper
|
|||||||||||||||||||||||||||||||||||||
|
3. Point-by-point response to Comments and Suggestions for Authors |
|||||||||||||||||||||||||||||||||||||
|
Comments 1: This very interesting manuscript is devoted to exploring PGE2 and LXA4 levels in migraine patients. It was shown, what PGE2 and LXA4 levels are significantly lower in migraine patients during both interictal and attack periods. Additionally, the levels of LXA4 and PGE2 decrease more with the prolongation of migraine attack duration. However, in the Discussion section there is no discussion of possible mechanisms and reasons for such reduced level. I think that it would be add to the Discussion chapter.
|
|
Response 1: Thank you for pointing this out. In line with your suggestions we've added a few more sentences to our discussion sentences regarding LXA4 and PGE2. The new versions are as follows; “LXA4 has been shown to have anti-inflammatory and neuroprotective effects in a va-riety of neurological diseases [22,23]. Mice lacking the ALX/FPR2 homolog develop a se-vere inflammatory response after reperfusion in cerebral ischemia [24]. LXA4 reduces neurogenic damage and promotes neuroprotection in ischemic stroke [24]. Studies in rats have shown that LXA4 injections into the ischemic brain tissue reduce stroke volume [20]. LXA4 also positively affects ischemic stroke by correcting the blood-brain barrier (BBB) dysfunction that develops after stroke [25,26]. Moreover, a recent study showed that SPMs, including LXA4 is reduced in multiple sclerosis (MS) patients [23]. Lipid metabolites in plasma patients of MS correlate with disease progression, and SPM production is dimin-ished in leukocytes [27]. The most commonly used molecules in acute and chronic pain are NSAIDs, opioids, and corticosteroids. However, their use has problems such as side effects, addiction, and tolerance [28]. Isolated SPMs have been used as analgesics in experimental models at very low doses without side effects [29]. Moreover, it has been shown that diets containing eicosapentaenoic acid and docosahexaenoic acid, which are precursors of SPMs, reduce the frequency and duration of migraine attacks [30]. LXA4 is endogenous SPM in all SPMs and has analgesic effects. Intravenous or intrathecal injections of LXA4 reduced thermal hyperalgesia in rats [31]. Miao et al. showed that LXA4 effectively reduced radicular pain in a rat non-compressive lumbar disc herniation model [32]. Martini et al. also revealed that LXA4 may efficiently regulate microglial activation and reduce neuropathic pain in animals following spinal cord hemisection [33]. In our study, LXA4 was significantly lower in migraine patients than in the healthy group during the attack and interictal pe-riods. This result is striking and, as far as we know, is not included in the literature. De-creased LXA4 levels suggested that migraine patients may have impaired LXA4 produc-tion or catabolism. However, the literature cannot provide sufficient information on this subject. LXA4 may be a new and promising agent, and has the potential to help resolve inflammation and shorten the duration of migraine attacks. While these findings are promising, additional research is needed to confirm their validity and to develop safe and effective LXA4-based therapies [34]. “ “PGs are signaling molecules produced by cyclooxygenase (COX) enzymes from AA. PGE2 is the most common prostaglandin in the human body. PGE2 receptors are a family of G protein-coupled receptors (GPCRs), a lipid mediator in inflammation, pain, and im-munity [35]. PGE2 levels have been linked to a variety of neurological diseases, including Alzheimer's disease, multiple sclerosis, and migraine. In Alzheimer's patients, PGE2 lev-els are high [35]. PGE2 appears to have a protective effect in multiple sclerosis by activat-ing myelin-producing oligodendrocytes [36]. PGE2 is also implicated in the mechanisms underlying migraine pain [37]. Several studies demonstrated that the infusion of PGE2 causes headaches in 83-100% of patients, but the headache occurrence rate after prosta-glandin F2-alpha infusion was just 17% [38-40]. Internal jugular venous blood levels of PGE2 have been found by Sarchielli et al. to peak two hours after the onset of a headache, remain throughout hours 4-6 post-headache start, and then drop [38]. PGE2 levels in mi-graine patients' saliva and nasal lavage samples taken during the interictal stage are sim-ilar to those in control subjects, according to research by Mohammadian and colleagues [41]. Tuca et al. found that PGE2 levels in migraine sufferers' saliva are considerably greater during a headache attack than in samples obtained between attacks [7]. A recent study analyzing COX-2 levels in migraine patients discovered a statistically significant upregulation of this PGE2-producing enzyme during headache episodes [42]. In our study, PGE2 levels have been demonstrated to be higher in migraine patients during the attack period than in the interictal period, which is consistent with previous research. Furthermore, it was significantly lower than the healthy control group during the attack and interictal periods. This finding could imply that the role of PGE2 in migraine pain and the pathophysiological pathways are more complex than previously understood. More detailed research is required on this topic.” |
|
Comments 2: In the chapter materials and methods are indicated, that after blood collection, Diclofenac sodium 75 mg was administered intramuscularly as a treatment. Were samples taken after treatment with Diclofenac to investigate PGE2 and LXA4 levels? |
|
Response 2: We used 75 mg Diclofenac sodium for migraine attack treatment. But we did not evaluate the patients immediately after treatment. We collected control blood samples from patients at least seven days later. Because we think that diclofenac treatment may affect PGE2 and LXA4 levels. Therefore, we waited at least 7 days for this affect to disappear. |
